# Impact of Urethral Sphincter Electrophysiology on Botulinum Toxin A Treatment in Women with Non-Neurogenic Dysfunctional Voiding

**DOI:** 10.3390/biomedicines12081902

**Published:** 2024-08-20

**Authors:** Tien-Lin Chang, Yuan-Hong Jiang, Hann-Chorng Kuo

**Affiliations:** Department of Urology, Hualien Tzu Chi Hospital, Buddhist Tzu Chi Medical Foundation and Tzu Chi University, Hualien 970, Taiwan

**Keywords:** electrophysiology study, electromyography, dysfunctional voiding, nerve conduction velocity

## Abstract

Dysfunctional voiding (DV) is an abnormal urethral sphincter activity during voiding in neurologically normal individuals. Urethral sphincter botulinum toxin A (BoNT-A) injection has been used to treat DV, but the results have not been completely satisfactory. This study investigated the neurological characteristics of women with DV using the lower urinary tract electrophysiology (EP) study and the therapeutic efficacy of BoNT-A injection. In total, 48 women with DV and 16 women with normal voiding were included. Videourodynamic studies were conducted to diagnose DV before BoNT-A injection. EP studies, including urethral sphincter electromyography, bulbocavernosus reflex, and pudendal nerve conduction velocity, were conducted. Polyphasic motor unit action potentials suggestive of reinnervation were detected in 58.3% of patients with DV and 18.8% of controls (*p* = 0.001). Significant improvement in the corrected maximum flow rate (cQmax) was observed in patients with reinnervation at 1 and 3 months after BoNT-A injections into the urethral sphincter. Urethral sphincter denervation or reinnervation activity was commonly noted in 62.5% of women with DV. Repeated BoNT-A injections into the urethral sphincter provided effective treatment in 47.9% of patients, with mild improvement in cQmax observed in patients with urethral sphincter reinnervation. However, the improvement was not superior to those without reinnervation.

## 1. Introduction

Dysfunctional voiding (DV) is an abnormality of bladder emptying in neurologically normal individuals with increased external sphincter activity during voluntary voiding, causing functional bladder outlet obstruction (BOO). DV can cause various lower urinary tract symptoms, including storage and emptying symptoms. DV is the most common form of voiding dysfunction in women with clinically unsuspected BOO [1]. Furthermore, it can cause recurrent urinary tract infections, acute or chronic urinary retention, and, in severe cases, upper and lower urinary tract decompensation [2]. Accurate diagnosis of DV is essential for selecting appropriate treatment. Videourodynamic studies (VUDS) can provide an accurate diagnosis of DV. The VUDS of DV is characterized by increased external sphincter activity during voiding, high voiding detrusor pressure (Pdet), low maximum flow rate (Qmax), and increased post-void residual (PVR) volume, as well as a typical downward spiral shape in voiding cystourethrography [3]. DV is often encountered in adult women. In a retrospective analysis of VUDS in 1914 women with voiding dysfunction, DV was detected in 325 (17%), and poor relaxation of the urethral sphincter was observed in 336 women (17.6%) [3]. Adult women with lower urinary tract symptoms may have a high prevalence of DV history during childhood [4].

Effective treatments for DV include biofeedback pelvic floor muscle exercises, antimuscarinic therapy, sacral neuromodulation, and posterior tibial nerve stimulation [5,6,7,8]. Since no definitive treatment has been reported for DV, external urethral sphincter (EUS) botulinum toxin A (BoNT-A) injection has been used to relax the EUS [9]. However, not all patients with DV have satisfactory treatment outcomes after BoNT-A injection [10]. Although enthusiastic bladder management has been provided to patients with DV, many still cannot resume normal micturition and clean intermittent catheterization is needed [11]. These unsuccessful treatment outcomes for DV may be due to an inadequate understanding of the underlying pathophysiology, such as detrusor dysfunction, urethral sphincter or pelvic floor dysfunction, or abnormal micturition reflex regulation [12]. Therefore, meticulous neurologic and urodynamic examinations are necessary to determine possible electrophysiological pathophysiology and appropriately select patients with specific DV subtypes for medical or surgical treatment. Thus, this study aimed to investigate the neurological characteristics of patients with DV using the lower urinary tract electrophysiology (EP) study and the therapeutic efficacy of BoNT-A injection into the urethral sphincter for DV. Based on the results of this study, we might have evidence to select appropriate patients with DV and certain EP characteristics for BoNT-A injection into the urethral sphincter.

## 2. Materials and Methods

A total of 48 women diagnosed with VUDS-confirmed DV at the department of urology of a single medical center between November 2020 and April 2023 were enrolled in this study. Women with DV were enrolled in this clinical trial for urethral BoNT-A treatment. Furthermore, 16 women without voiding dysfunction were enrolled as controls. Because this was a preliminary observational study, no power calculation was made in the study design. This study was approved by the Institutional Review Board of the hospital (approval number: 110-265-A). Informed consent was obtained from all patients.

All patients underwent VUDS for DV diagnosis. The VUDS procedure and reported parameters were in accordance with the recommendations of the International Continence Society [13]. In VUDS, DV is characterized by an intermittent and/or fluctuating flow rate due to involuntary intermittent contractions of the periurethral striated or levator muscles during voiding in neurologically normal women [14]. Patients with a voiding detrusor pressure >35 cmH_2_O, Qmax < 15 cmH_2_O, spinning top appearance of urethral narrowing during voiding in VUDS, and symptoms of lower urinary tract dysfunction, including severe difficult urination, large PVR volumes, and chronic urinary retention, were diagnosed with DV and included in this study [1,10]. Patients with a voiding detrusor pressure <35 cmH_2_O, those without a narrow urethra during voiding, those without non-neurogenic pelvic floor dysfunction, and those with overt neurogenic lower urinary tract dysfunction, such as intracranial lesions, spinal cord injury, myelomeningocele, multiple sclerosis, and transverse myelitis, were excluded from this study.

Patients underwent lower urinary tract EP studies, including the bulbocavernosus reflex (BCR) by electrical stimulation, nerve conduction velocity (NCV) study of the pudendal nerve, and concentric electromyography (EMG) study of the EUS using Medtronic Keypoint EP equipment [15]. EP studies were conducted before the first BoNT-A injection into the urethral sphincter, with intravenous sedation, with the patients placed in the dorsal lithotomy position. EP testing of the BCR was performed using external stimulation of the dorsal nerve of the clitoris, and signals were recorded using surface electrodes on the perianal striated sphincter muscle. Thus, an objective study of the sacral reflex arc (S2–S4) was conducted. The latency time in a normal BCR is approximately 33 ms, and prolongation of this interval is a sign of a pathological reflex. An NCV study of the pudendal nerve was performed using a St. Mark’s electrode to transrectally stimulate the bilateral pudendal nerves at the level of the ischial spine and to record the response at the external anal sphincter muscle [16]. The typical pudendal nerve terminal motor latency response in healthy individuals is approximately 2.5 ms with an amplitude of 1 mV. The concentric needle EMG study was conducted by inserting the EMG needle from the periurethral area. EMG is a diagnostic test to assess the neuromuscular function of muscles. The urethral sphincter muscles are tonically active and exhibit normal continuously firing motor unit action potentials (MUAPs) even at rest. For adequate analysis, at least 10 MUAPs should be recorded. Denervated muscle fibers may produce rhythmic spontaneous electric potentials, such as fibrillation and positive sharp waves, which are thought to be characteristic of denervation changes. The presence of polyphasic or giant MUAPs is a sign of reinnervation [17] (Figure 1). The urethral sphincter EMG study using a concentric needle has been widely used in detecting inadequate relaxation of the urethral sphincter in DV or predicting treatment outcome of sacral neuromodulation in patients with Fowler’s syndrome, with good reliability and validity [18,19].

BoNT-A injections into the urethral sphincter were performed in the operating room for all patients under intravenous general anesthesia. Patients were admitted to the hospital for a 100 U BoNT-A injection every 3 months, and the urethral sphincter BoNT-A injection was performed four times. BoNT-A injection was performed based on a previous study [9]. After BoNT-A injections, a 14-Fr Foley catheter was routinely placed overnight. Patients were monitored at the outpatient clinic 1, 3, 6, and 9 months after the fourth BoNT-A injection. Uroflowmetry, maximum flow rate (Qmax), PVR, corrected Qmax (cQmax, Qmax divided by the square root of voided volume plus PVR), voiding efficiency, and the global response assessment (GRA) questionnaire for assessing voiding difficulty (scored from −3 (markedly worse) to +3 (markedly improved)), and visual analog scale (VAS) for dysuria (scored from 0 (no dysuria) to 10 (severe dysuria)) was performed to evaluate the treatment outcomes of urethral sphincter BoNT-A injections. Patients with improvement in cQmax and GRA were considered to have a satisfactory response to urethral BoNT-A injections.

Continuous variables were presented as mean ± standard deviation, and the rate of abnormal EP findings was presented. Differences in EP study parameters between the two groups were analyzed using the chi-square test. All calculations were performed using SPSS for Windows, version 16.0 (SPSS). A *p*-value < 0.05 was considered significant.

## 3. Results

A total of 64 women, including 48 with DV and 16 controls, with mean ages of 55.2 ± 16.1 and 61.1 ± 14.7 years, respectively, were included in this study. The controls were women who had stress urinary incontinence without voiding difficulty and were ready for anti-incontinence surgery. Table 1 presents the baseline demographics and VUDS diagnosis of patients with DV. Of the 48 patients, 7 (14.6%) had undergone transurethral incision of the bladder neck (TUI-BN) for previously diagnosed bladder neck dysfunction, but DV was present after TUI-BN. Additionally, 10 (20.8%) patients had previous hysterectomy, and 4 (8.3%) had previous spinal surgery. Among the patients with DV, 38 (79.2%) had concomitant urodynamic detrusor overactivity (DO), 1 (2.1%) had detrusor underactivity (DU), and 2 had vesicoureteral reflux during the VUDS examination. Of the 16 controls, 4 (25%) had intrinsic sphincter deficiency, 2 (12.5%) had DU, and 3 (18.8%) had DO.

Table 2 presents the VUDS and EP study parameters of patients with DV and controls. In the VUDS, no significant differences were observed in Qmax, voided volume, PVR, cystometric bladder capacity, corrected Qmax (cQmax), voiding efficiency, or bladder contractility index. The voiding detrusor pressure was significantly higher in patients with DV than in controls. In the EP study, 60.4% of patients with DV and 87.5% of controls had a normal BCR latency time (*p* = 0.092). In the NCV study, decreased amplitude was observed in 75.0% of patients with DV and 56.2% of controls. In the EMG studies, only two patients with DV (4.2%) had positive sharp waves, indicating denervation, whereas the controls showed no positive sharp waves. Polyphasic MUAPs were observed in 58.3% of patients with DV who presented with reinnervation, whereas they were observed only in 18.8% of controls (*p* = 0.001).

The 48 women with DV were divided into two subgroups: those with reinnervation (*n* = 28) and those without reinnervation (including denervation and normal EMG subgroup, *n* = 20). Table 3 presents the VUDS and EP study parameters. There were no significant differences in the baseline VUDS and EP parameters between the two subgroups. However, after BoNT-A injections into the urethral sphincter, cQmax significantly increased at 1 and 3 months after BoNT-A treatment in patients with DV with reinnervation, whereas it did not increase in patients without reinnervation (Table 4). However, no significant differences in GRA results were observed between the two subgroups at 1, 3, 6, and 9 months after BoNT-A injections into the urethral sphincter. Of the 48 patients with DV, 23 (47.9%), including 13 (46.4%) and 10 (50.0%) with and without reinnervation, respectively, reported satisfactory treatment outcomes (GRA ≥ 2) after urethral sphincter BoNT-A injections (*p* = 0.503).

Table 5 presents the characteristics of patients with GRA ≥ 2 and GRA < 2 after BoNT-A injection into the urethral sphincter. No significant differences in VUDS and EP parameters were observed between the two subgroups. The VAS score for dysuria significantly improved at all time points after BoNT-A treatment in both subgroups. However, no significant difference in the improvement of the VAS score for dysuria was observed between the two groups.

## 4. Discussion

This study showed that 62.5% of women with a videourodynamic diagnosis of non-neurogenic DV and dysuria had neurological deficits in the urethral sphincter in the EP study. After repeated urethral sphincter BoNT-A injections, only 47.9% of patients with DV had satisfactory treatment outcomes. Patients with DV and reinnervation in the EP study showed a significant improvement in cQmax at 1 and 3 months after BoNT-A injection, and the therapeutic effect gradually declined. However, the changes in GRA and improvement in VAS score for dysuria were not significant compared with patients with DV without reinnervation.

The actual pathophysiology of DV has not been elucidated. DV might be associated with lower urinary tract neuropathy, including BCR reflex arc, pudendal neuropathy, and urethral sphincter neuropathy [17]. Potential neurological deficits should be considered before treatment. In our previous EP study, a high percentage of neurological deficiency was observed in patients with DU [15]. The BCR, pudendal NCV study and urethral sphincter EMG were used to assess lower urinary tract dysfunction and neuropathy. The BCR test is used to evaluate whether the conus medullaris and the S2–S4 pelvic nerves are intact. Testing for the absence of this reflex may indicate neurological deficits located at the pelvic nerves or cauda equina [16]. The pudendal NCV study tests pudendal nerve injury or other neuropathy. A prolonged latency time or lower nerve conduction velocity indicates possible nerve demyelination changes, whereas decreased amplitude in the pudendal NCV study indicates axonopathy [20]. The motor-unit EMG assesses the neuromuscular function of the urethral sphincter using concentric needle EMG [21]. Abnormal polyphasic high-amplitude MUAPs have been observed in female voiding dysfunction due to DV or Fowler’s syndrome [22]. The high rate of reinnervation observed in EMG studies of EUS indicates that urethral sphincter neuropathy occurs, and abnormally increased urethral sphincter activity is present in most patients with DV [23].

In this study, the absence of BCR, decreased amplitude in pudendal NCV studies, and changes in urethral sphincter reinnervation were detected in 39.6%, 75.0%, and 58.3% of patients with DV, respectively. All of the incidences of abnormal EP studies were higher in patients with DV than in the control group, even though only the reinnervation changes were statistically significant (*p* = 0.001). Reinnervation of the urethral sphincter indicates incomplete or inadequate recovery from previous neurological insults, which may result in a non-relaxing urethral sphincter (in denervation DV) or impaired relaxation of the urethral sphincter (in reinnervation DV) during voiding, leading to functional BOO and voiding difficulty [24]. Furthermore, the absence of BCR and decreased amplitude in the pudendal NCV indicates neuropathy of the micturition reflex arc or internal pudendal nerve.

DV is defined as the habitual contraction/hyperactivity of the urethral sphincter during voiding, resulting in high voiding pressure with a low Qmax and a spinning top appearance on voiding cystourethrography [25]. DV causes voiding symptoms, such as a slow stream and large PVR volume. Attempts to reduce hypertonicity or hyperactivity of the urethral sphincter with medication and resume spontaneous voiding usually result in treatment failure [11]. Psychological factors contributing to voiding dysfunction, such as anxiety and depression, have been suggested to cause low detrusor contractility and non-relaxing urethral sphincter by inhibiting detrusor contraction [26]. Although urethral sphincter BoNT-A injections can reduce urethral resistance, the decrease in urethral sphincter activity might not have a pharmacological effect on DV due to psychological insult [26].

Treatment options for DV in women and children include psychological support, medical therapy, biofeedback physiotherapy, urethral sphincter BoNT-A injection, sacral neuromodulation, and percutaneous tibial nerve stimulation [27,28]. Previous studies have reported a successful treatment outcome rate of 60%–100% in patients with DV, including pediatric and adult patients, after urethral sphincter BoNT-A injections [29,30,31]. Patients with successful treatment outcomes showed subjective improvement in dysuria and improvement in voiding pressure and PVR. In this study, patients with DV showed improved cQmax in the first and third months after urethral sphincter BoNT-A injections. Furthermore, VAS scores for dysuria improved in the first, third, sixth, and ninth months after BoNT-A treatment. These findings indicated that urethral sphincter BoNT-A injections reduced urethral resistance. However, the therapeutic effects were limited in patients with DV with reinnervation. Although the initial therapeutic effect might be satisfactory, the effect declined gradually with time. Therefore, repeated urethral sphincter BoNT-A injections are needed to maintain therapeutic efficacy.

The therapeutic mechanism of BoNT-A in striated or smooth muscle involves the inhibition of acetylcholine release from nerve terminals, resulting in the paralysis of part of the muscle cells [10,32]. After injecting BoNT-A into the urethral sphincter, it relaxes, and BoNT-A may be transported into the dorsal horn of the sacral cords, inhibiting the chronic inflammation that causes hyperactivity of the urethral sphincter and providing dual mechanisms of action that result in improving voiding condition [33]. However, only approximately half of patients with DV can have a good benefit from urethral sphincter BoNT-A injection [10,34,35]. Patients with acontractile detrusor and urinary retention could remove a catheter and reduce PVR after urethral BoNT-A injection [36]. An analysis of patients’ baseline characteristics revealed that patients with a higher baseline BOO index had a successful response to BoNT-A injection [35]. In women with a more severe form of DV, Fowler’s syndrome, the urethral BoNT-A injection improved symptoms after treatment [37]. This study showed that the characteristics of DV in VUDS, higher voiding pressure, and reinnervation EMG subtypes did not play a role in predicting successful treatment outcomes. The usual dose of BoNT-A injection for adult patients with DV is 100 U [9,10], which might be inadequate to paralyze the urethral sphincter or effective in inhibiting sacral cord inflammation. It is also possible that we do not fully understand the underlying pathophysiology of DV [38]. Other unknown pathophysiological factors underlying DV may also contribute to this unsatisfactory treatment outcome and warrant further investigation.

The pathophysiology of DV in neurologically normal women is not well elucidated. Although neurological deficits were observed in this study, the reason why BoNT-A injection could not effectively relax the urethral sphincter remains unknown. Our previous studies on urinary biomarkers in women with lower urinary tract dysfunction have shown that women with DV had higher urinary levels of tumor necrosis factor-alpha and 8-hydroxydeoxyguanosine, and urinary interleukin-2 levels were significantly lower [39]. These findings indicate that chronic bladder inflammation might also play a role in DV, resulting in increased sensory activity and hyperactivity in the urethral sphincter. Future studies must focus on the interaction between urinary bladder and urethral sphincter activity, and the treatment should involve both dysfunctions.

This study has several limitations. First, the EP study technique may have some errors due to inexperienced hands. In this study, the absence of BCR was observed in a high percentage of patients with DV and some controls. Second, this study included women with stress urinary incontinence as controls. Although these women did not have voiding dysfunction, pudendal nerve injury due to multiple pregnancies and delivery might have resulted in abnormal EP results [40]. Furthermore, data on the recruitment of urethral sphincter EMG could not be assessed because the EP study was conducted before lower urinary tract surgery under intravenous general anesthesia. These confounding factors might have affected the results of this study. Nevertheless, the results showed that a high percentage of patients with DV had reinnervation or denervation of the urethral sphincter. Finally, the purpose of this study was to investigate the effect of neurological characteristics of DV on the therapeutic efficacy of BoNT-A injection. The data on therapeutic results of BoNT-A were not included in the results section. The electrophysiological condition of the urethral sphincter was not expected to change after the BoNT-A injection. Therefore, repeated examination was not performed after BoNT-A treatment. The findings of this study provide valuable insight into the pathophysiology of DV and can help determine the optimal treatment strategy.

## 5. Conclusions

This study showed that 62.5% of women with DV had urethral sphincter reinnervation or denervation. After repeated BoNT-A injections into the urethral sphincter, 47.9% of patients had satisfactory treatment outcomes. Furthermore, the results showed that higher voiding detrusor pressure and DV subtypes in VUDS were not predictors of treatment outcomes. Patients with DV, along with urethral reinnervation, were found to have improved cQmax after repeated urethral sphincter BoNT-A injections during follow-up.

## Figures and Tables

**Figure 1 biomedicines-12-01902-f001:**
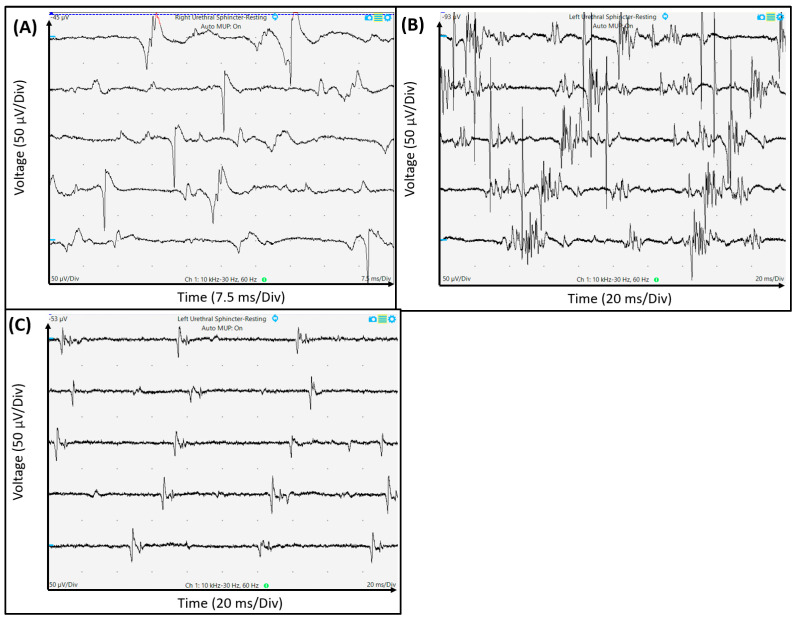
Different electromyographic findings in patients with dysfunctional voiding. (**A**) Positive sharp waves and the presence of fibrillation indicating denervation changes. (**B**) Polyphasic motor-unit action potential indicating reinnervation changes. (**C**) Normal motor unit action potentials. X-labeling: time (millisecond), Y-labeling: Voltage, Div: Digital input group voltage.

**Table 1 biomedicines-12-01902-t001:** Demographic and VUDS findings of patients with dysfunctional voiding (*n* = 48).

Age (Years)	55.2 ± 16.1
Comorbidity	
Diabetes mellitus	7 (14.6%)
Hypertension	14 (29.2%)
Coronary artery disease	3 (6.3%)
Chronic kidney disease	2 (4.2%)
Recurrent urinary tract infection	7 (14.6%)
Previous surgery	
Transurethral incision of the bladder neck	7 (14.6%)
Hysterectomy	10 (20.8%)
Spinal surgery	4 (8.3%)
Videourodynamic study	
Detrusor overactivity	38 (79.2%)
Detrusor underactivity	1 (2.1%)
Vesicoureteral reflux	2 (4.2%)

**Table 2 biomedicines-12-01902-t002:** VUDS parameters and EP study data between patients with DV and controls.

	DV (*n* = 48)	Control (*n* = 16)	*p*-Value
Age (years)	55.2 ± 16.1	61.1 ± 14.7	0.184
VUDS parameters			
Qmax (mL/s)	9.3 ± 5.4	10.1 ± 15.0	0.520
Volume (mL)	174.3 ± 93.5	159.6 ± 202.7	0.950
PVR (mL)	156.5 ± 191.7	159.5 ± 198.1	0.960
CBC (mL)	310.7 ± 184.8	319.1 ± 194.5	0.883
cQmax	0.63 ± 0.42	0.65 ± 0.68	0.944
VE	0.57 ± 0.36	0.55 ± 0.46	0.842
Pdet (cmH_2_O)	50.9 ± 26.3	20.5 ± 18.0	0.001
BCI	96.0 ± 32.7	74.1 ± 74.8	0.263
EP parameters			
BCR latency time (ms)	1.78 ± 2.13	1.00 ± 1.41	0.185
Absence of BCR	19 (39.6%)	2 (12.5%)	0.092
L’t NCV (ms)	1.50 ± 1.57	1.05 ± 1.20	0.340
R’t NCV (ms)	2.03 ± 2.19	1.81 ± 1.84	0.548
Latency < 2.5 ms	33 (68.8%)	11 (68.8%)	0.396
L’t NCV amp. (mV)	2.19 ± 6.26	2.44 ± 4.07	0.881
R’t NCV amp. (mV)	1.32 ± 2.51	2.23 ± 5.32	0.393
Decreased amplitude	36 (75.0%)	9 (56.2%)	0.140
EMG			
Denervation	2 (4.2%)	0 (0.0%)	0.436
Reinnervation	28 (58.3%)	3 (18.8%)	0.001
Normal EMG	18 (37.5%)	13 (71.2%)	0.024

DV: dysfunctional voiding; Qmax: maximum flow rate; PVR: post-void residual volume; CBC: cystometric bladder capacity; cQmax: correct maximum flow rate (defined as Qmax/Volume1/2); VE: voiding efficiency; Pdet: voiding detrusor pressure at maximum urinary flow rate; BCI: bladder contractility index; EP: electrophysiological study; BCR: bulbocavernous reflex; NCV: nerve conduction velocity, amp: amplitude, EMG: electromyography.

**Table 3 biomedicines-12-01902-t003:** Baseline videourodynamic and electrophysiological study parameters between patients of dysfunctional voiding with or without reinnervation of the urethral sphincter.

	DV with ReIN (*n* = 28)	DV without ReIN (*n* = 20)	*p*-Value
Age (years)	56.7 ± 15.3	53.4 ± 17.9	0.528
VUDS parameters			
Qmax (mL/s)	9.3 ± 5.4	9.4 ± 6.1	0.981
Volume (mL)	164.7 ± 78.5	164.2 ± 90.8	0.983
PVR (mL)	81.0 ± 163.4	137.2 ± 177.6	0.293
CBC (mL)	239.8 ± 180.2	301.4 ± 158.9	0.262
cQmax	0.68 ± 0.45	0.61 ± 0.41	0.606
VE	0.79 ± 0.27	0.67 ± 0.37	0.252
Pdet (cmH_2_O)	49.3 ± 26.1	47.9 ± 23.4	0.859
BCI	94.3 ± 36.5	94.8 ± 22.9	0.964
EP parameters			
BCR latency (ms)	1.60 ± 1.53	2.57 ± 2.93	0.279
Absent BCR	12 (42.9%)	10 (50.0%)	0.444
L’t NCV latency (ms)	1.85 ± 1.87	1.07 ± 0.70	0.128
R’t NCV latency (ms)	2.03 ± 1.62	2.24 ± 3.16	0.765
Latency < 2.5 ms	14 (50.0%)	8 (40.0%)	0.745
L’t NCV amp. (mV)	1.96 ± 7.08	1.43 ± 2.42	0.782
R’t NCV amp. (mV)	0.75 ± 1.66	1.12 ± 1.49	0.480
Decreased amplitude	25 (89.3%)	14 (70.0%)	0.647

DV: dysfunctional voiding; ReIN: reinnervation; Qmax: maximum flow rate; PVR: post-void residual volume; CBC: cystometric bladder capacity; cQmax: correct maximum flow rate (defined as Qmax/Volume1/2); VE: voiding efficiency; Pdet: voiding detrusor pressure at maximum urinary flow rate; BCI: bladder contractility index; BCR: bulbocavernous reflex; NCV: nerve conduction velocity.

**Table 4 biomedicines-12-01902-t004:** Comparison of postoperative cQmax between patients with DV with or without reinnervation or denervation and GRA.

	DV with ReIN (*n* = 28)	DV without ReIN (*n* = 20)	*p*-Value
Pre-OP	0.73 ± 0.44	0.60 ± 0.38	0.733
Post-OP 1M	1.10 ± 0.68 *	0.70 ± 0.41	0.298
Post-OP 3M	1.03 ± 0.56 *	0.61 ± 0.30	0.314
Post-OP 6M	0.77 ± 0.32	0.55 ± 0.68	0.481
Post-OP 9M	0.61 ± 0.78	0.43 ± 0.51	0.308
GRA at 1 month	1.98 ± 1.44	1.77 ± 0.23	0.834
GRA at 3 months	1.69 ± 0.64	1.66 ± 1.48	0.733
GRA at 6 months	1.34 ± 1.89	1.03 ± 2.67	0.246
GRA at 9 months	1.19 ± 1.02	1.36 ± 0.92	0.641

GRA: global response assessment; DV: dysfunctional voiding; cQmax: correct maximum flow rate (defined as Qmax/Volume1/2); ReIN: reinnervation; * Significant difference compared with pre-op data (*p* < 0.05).

**Table 5 biomedicines-12-01902-t005:** Baseline VUDS and EP parameters of patients with DV with GRA ≥ 2 and GRA < 2 after urethral sphincter BoNT-A injections.

	GRA ≥ 2 (*n* = 23)	GRA < 2 (*n* = 25)	*p*-Value
Age (years)	56.9 ± 18.0	65.2 ± 19.8	0.457
VUDS parameters			
Qmax (mL/s)	7.7 ± 3.9	8.7 ± 5.9	0.600
Volume (mL)	150.8 ± 83.1	169.8 ± 121.1	0.599
PVR (mL)	158.8 ± 196.3	155.0 ± 192.8	0.953
CBC (mL)	300.1 ± 167.9	317.8 ± 198.5	0.772
cQmax	0.62 ± 0.32	0.68 ± 0.37	0.729
VE	0.58 ± 0.33	0.57 ± 0.38	0.972
Pdet (cmH_2_O)	57.5 ± 38.0	50.8 ± 28.8	0.539
BCI	100.0 ± 39.6	88.0 ± 46.1	0.410
EP parameters			
BCR latency (ms)	1.50 ± 2.87	2.13 ± 1.90	0.493
BCR < 33 ms	10 (43.5%)	16 (64.0%)	0.500
L’t NCV latency (ms)	1.28 ± 1.13	1.64 ± 1.98	0.511
R’t NCV latency (ms)	1.88 ± 1.65	2.45 ± 2.82	0.477
Latency < 2.5 ms	9 (39.1%)	13(52.0%)	1.000
L’t NCV amp. (mV)	3.24 ± 8.08	0.74 ± 1.22	0.228
R’t NCV amp. (mV)	1.85 ± 3.09	0.60 ± 1.05	0.089
Amplitude > 1 mV	4 (17.4%)	6 (24.0%)	0.706
Reinnervation EMG	13 (56.5%)	15 (60.0%)	0.503
Dysuria VAS			
at 1 month	3.83 ± 1.58 *	4.18 ± 2.94 *	0.234
at 3 months	3.34 ± 2.49 *	2.98 ± 2.88 *	0.568
at 6 months	2.47 ± 1.83 *	2.58 ± 2.67 *	0.417
at 9 months	2.52 ± 2.54 *	2.77 ± 2.33 *	0.453

GRA: global response assessment; DV: dysfunctional voiding; Qmax: maximum flow rate; PVR: post-void residual volume; CBC: cystometric bladder capacity; cQmax: correct maximum flow rate (defined as Qmax/Volume1/2); VE: voiding efficiency; Pdet: voiding detrusor pressure at maximum urinary flow rate; BCI: bladder contractility index; BCR: bulbocavernosus reflex; NCV: nerve conduction velocity; VAS: visual analog scale; * Significant difference compared with pre-op data (*p* < 0.05).

## Data Availability

The original contributions presented in the study are included in the article, further inquiries can be directed to the corresponding author.

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
