# Peer review of "Impact of Urethral Sphincter Electrophysiology on Botulinum Toxin A Treatment in Women with Non-Neurogenic Dysfunctional Voiding"

_biomedicines, 2024, doi:10.3390/biomedicines12081902_

Round 1

Reviewer 1 Report

Comments and Suggestions for Authors

Dear authors,

Thank you for your creative manuscript. However, there are several issues to be discussed before a possible publication.

- you used VUDS and EMG in order to evaluate VD but you only used subjective tools in order to estimate your treatment outcomes. This could be regarded as a separate limitation of your study and you have to discuss it in detail.

- why didn't you perform a repeat objective evaluation during your monitoring? Please, explain in the discussion section.

- what about those women with non-neurogenic pelvic floor disfunction? This could be another cause of voiding dysfunction which could not response in urethral sphincter BONT/A.

Author Response

Reviewer #1

Thank you for your creative manuscript. However, there are several issues to be discussed before a possible publication.

- you used VUDS and EMG in order to evaluate VD but you only used subjective tools in order to estimate your treatment outcomes. This could be regarded as a separate limitation of your study and you have to discuss it in detail.

Reply: Thank you for the comment. Indeed, the treatment outcome should be evaluated by more detailed examination, such as voiding pressure, voiding efficiency, and bladder outlet obstruction index. We have published several articles to report the therapeutic efficacy of urethral BoNT-A injections on female dysfunctional voiding (DV). [Ref. 10,35,36] In this study, these post-treatment assessments were also performed. (Lines 150-133) However, because the satisfaction rated by the patients is rather subjective, therefore, we used corrected maximum flow rate (Qmax) (Qmax divided by the square root of voided volume plus PVR) and global response assessment (GRA) to evaluate the effect of urethral sphincter electrophysiology on treatment outcome of BoNT-A injection. (Lines 156-157) Because the purpose of this study is to investigate the effect of neurological characteristics of DV on therapeutic efficacy of BoNT-A injection. The data of therapeutic results of BoNT-A are not put in the result section. We will add this statement in the limitation of the study. (Lines 321-324)

- why didn't you perform a repeat objective evaluation during your monitoring? Please, explain in the discussion section.

Reply: Thank you for the comment. The purpose of the electrophysiology study of urethral sphincter activity aims to investigate the denervation and reinnervation status of patients with a hyperactive urethral sphincter or non-relaxing urethral sphincter during voiding. The electrophysiology condition was supposed not change after BoNT-A injection. Therefore, we did not perform a repeat examination after BoNT-A treatment. We have added this statement to the Discussion section. (Lines 324-326)

- what about those women with non-neurogenic pelvic floor disfunction? This could be another cause of voiding dysfunction which could not response in urethral sphincter BONT/A.

Reply: Thank you for the comment. Patients with non-neurogenic pelvic floor dysfunction usually present with similar lower urinary tract symptoms such as dysuria, large PVR, or urinary retention, however, the video urodynamic study finding is distinct with that in dysfunctional voiding.  In this study, we only included patients with a voiding detrusor pressure >35 cmH2O, Qmax <15 cmH2O, spinning top appearance of urethral narrowing during voiding in VUDS, and symptoms of lower urinary tract dysfunction, including severe difficult urination, large PVR volumes, and chronic urinary retention, who were diagnosed with DV. (Lines 107-111) Patients with non-neurogenic pelvic floor dysfunction were not included. (Line 112)

Reviewer 2 Report

Comments and Suggestions for Authors

General comment

The manuscript entitled “Urethral Sphincter Electrophysiology Study and Impact on Botulinum Toxin A Treatment in Women with NonNeurogenic Dysfunctional Voiding” provides valuable insights into the neurological underpinnings and treatment outcomes of dysfunctional voiding using urethral sphincter botulinum toxin A injections. The study investigates lower urinary tract dysfunction through electrophysiological studies and evaluates the efficacy of botulinum toxin A injections in improving clinical outcomes. While the study addresses an important clinical challenge, several aspects require clarification and improvement before publication.

INTRODUCTION

The introduction effectively sets the stage by highlighting the lack of definitive medical treatments for dysfunctional voiding and the rationale for exploring botulinum injections as a therapeutic option. However, it would benefit from a clearer statement of the study's objectives and hypotheses. Specifically, articulating the gap in current knowledge and the specific contributions of the study would enhance its impact.

Provide a brief epidemiological data of dysfunctional voiding, also reporting risk factors for this issue. To this regard, see: 10.4081/aiua.2022.1.46 and 10.5489/cuaj.5230

METHODS

Provide the rationale for selecting 48 women with dysfunctional voiding and only 16 controls in terms of statistical power and representativeness.

The description of electrophysiology study methods is thorough but additional information on the reliability and validity of the techniques would strengthen the methodological rigor.

DISCUSSION

Address the limitations of the current study

Discuss alternative explanation for findings, including confounding variables and methodological limitations

Providing recommendations for future research directions

Comments on the Quality of English Language

The manuscript adheres to scientific writing conventions but would benefit from proofreading for grammatical errors and clarity of expression.

Author Response

Dear Reviewer:

Thank you for the constructive comments and suggestions. The followings are the point-to-point replies to individual comment.

Reviewer #2

General comment

The manuscript entitled “Urethral Sphincter Electrophysiology Study and Impact on Botulinum Toxin A Treatment in Women with Non-Neurogenic Dysfunctional Voiding” provides valuable insights into the neurological underpinnings and treatment outcomes of dysfunctional voiding using urethral sphincter botulinum toxin A injections. The study investigates lower urinary tract dysfunction through electrophysiological studies and evaluates the efficacy of botulinum toxin A injections in improving clinical outcomes. While the study addresses an important clinical challenge, several aspects require clarification and improvement before publication.

INTRODUCTION

The introduction effectively sets the stage by highlighting the lack of definitive medical treatments for dysfunctional voiding and the rationale for exploring botulinum injections as a therapeutic option. However, it would benefit from a clearer statement of the study's objectives and hypotheses. Specifically, articulating the gap in current knowledge and the specific contributions of the study would enhance its impact.

Reply: Thank you for the comment. In the second paragraph of Introduction, we have highlighted the need for a diagnostic test to determining appropriate patients with DV and certain electrophysiological characteristics for urethral BoNT-A injection. (Lines 79-85) At the end of this paragraph, we also added a statement for the study’s objectives and hypothesis. (Lines 87-89)

Provide a brief epidemiological data of dysfunctional voiding, also reporting risk factors for this issue. To this regard, see: 10.4081/aiua.2022.1.46 and 10.5489/cuaj.5230

Reply: Thank you for the comment. We have added a statement to report the epidemiological data and the relationship with DV in childhood and adult women. References are also added.

DV is not infrequently encountered in adult women. In a retrospective analysis of VUDS in 1914 women with voiding dysfunction, DV was detected in 325 (17%) and poor relaxation of urethral sphincter in 336 (17.6%). [3] Adult women with lower urinary tract symptoms may have a high prevalence of DV history in her childhood. [4] (Lines 67-70)

METHODS

Provide the rationale for selecting 48 women with dysfunctional voiding and only 16 controls in terms of statistical power and representativeness.

Reply: Thank you for the comment. The women with DV were consecutively entered clinical trial for urethral BoNT-A treatment (IRB: 110-265-A). (Lines 95-96) Because this was a preliminary observational study to investigate the neurological characteristics of patients with DV using the lower urinary tract electrophysiology study and the therapeutic efficacy of urethral sphincter BoNT-A injection for DV. No power calculation was made in the study design. (Lines 97-98)

The description of electrophysiology study methods is thorough but additional information on the reliability and validity of the techniques would strengthen the methodological rigor.

Reply: Thank you for the comment. We have added statement to address the clinical usefulness and validity of urethral sphincter EMG study. (Lines 139-142)

DISCUSSION

Address the limitations of the current study.

Reply: Thank you for the comment. We have addressed the limitation of the study, including the suggestions from Reviewer #1, and the lack of repeat examination after BoNT-A treatment. (Lines 321-326)

Discuss alternative explanation for findings, including confounding variables and methodological limitations

Reply: Thank you for the comment. We have discussed the findings of this study, (Lines 235-237, and Lines 241-250), and the possible mechanism for the low successful treatment result in this study. (Lines 291-295)

In addition, we have addressed several limitations of the study, including the EP study was performed under general anesthesia, (Lines 316-318) the control women were stress urinary incontinence and pudendal nerve injury might exist in these multiple delivery women. (Lines 313-316) These confounding factors might also affect the results of this study. (Lines 318-319)

Providing recommendations for future research directions

Reply: Thank you for the comment. We have added recommendations for future research direction. (Lines 300-309)

Round 2

Reviewer 2 Report

Comments and Suggestions for Authors

No further corrections required.

Comments on the Quality of English Language

Minor grammar checks

Author Response

Dear Reviewer:

Thank you for the comment.

The grammatical errors in the manuscript has been appropriately corrected.